# Cascade Parallel Random Forest Algorithm for Predicting Rice Diseases in Big Data Analysis

**Lei Zhang [1], Lun Xie [1,\*], Zhiliang Wang [1] and Chen Huang [2]**

1   School of Computer and Communication Engineering, University of Science and Technology Beijing, Beijing 100083, China; b20150296@xs.ustb.edu.cn (L.Z.); wzl@ustb.edu.cn (Z.W.)
2   Taiji Computer Corporation Limited, Beijing 100083, China; huang@mail.taiji.com.cn
\*   Correspondence: xielun@ustb.edu.cn; Tel.: +86-01062332873

**Abstract:** Experts in agriculture have conducted considerable work on rice plant protection. However, in-depth exploration of the plant disease problem has not been performed. In this paper, we find the trend of rice diseases by using the cascade parallel random forest (CPRF) algorithm on the basis of relevant data analysis in the recent 20 years. To confront the problems of high dimensions and imbalanced data distributions in agricultural data. The proposed method diminishes the dimensions and the negative effect of imbalanced data by cascading several random forests. For experimental evaluation, we utilize the Spark platform to analyze botanic data from several provinces of China in the past 20 years. Results for the CPRF model of plant diseases that affect rice yield, as well as results for samples by using random forest, CRF, and Spark-MLRF are presented, and the accuracy of CPRF is 96.253%, which is higher than that of the other algorithms. These results indicate that the CPRF and the utilization of big data analysis are beneficial in solving the problem of plant diseases.

**Keywords:** predictive accuracy; big data analysis; agricultural plant disease; cascade parallel random forest algorithm

## 1. Introduction

### 1.1. Research Background and Meaning

The yield of agricultural products is considerably important for every country [1]. The Food and Agriculture Organization of the United Nations (FAO) predicted that the demand for agricultural production will be increased by 70% to sustain the subsistence of 9 billion people in 2050 [2]. As one of the most important major food sources, rice sustains more than 50% of worldwide people and significantly contributed to global food security [3]. Therefore, rice yield has a great impact on the economy and politics. Food shortages hinder economic development and even lead to social unrest; thus, food security is very important for developing countries [4]. On the other hand, although developed countries have abundant agricultural product resources, the agricultural plant protection should also be taken seriously [5], including crops yield prediction and botany disease prevention. Therefore, it is significant to predict rice yield and diseases to maintain the crop yield and ensure food security.

The analysis of the agricultural plant diseases that influence rice yields has motivated the study of cascade parallel random algorithm based on big data. The search for agricultural botany protection has become one of the widest computational problems in the big data field. Plant diseases, such as rice planthopper, bacterial leaf blight, brown spot, leaf smut, and rice leaf roller, are adverse to crop growth. The extent of plant diseases and insect pests is related to the output of agricultural products, such that data on plant diseases and insect pests should be efficiently dealt with urgently.

*1.2. Research Gap*

The problems of high-dimensional and imbalanced data cannot be solved well in the agricultural field [6]. On the one hand, the factors which influence the rice yield are discrete and highly dimensional. The weights of influencing factors are complicated. On the other hand, class imbalance problems usually exist in agricultural dataset.

*1.3. Contribution*

In our work, a cascade PRF (CPRF) algorithm is employed and deployed in big data platform for solving the problems of rice diseases detection and rice yield prediction.

The proposed CPRF algorithm combines the advantages of task parallelism and data cascade. For high dimensional and imbalanced agricultural data, CPRF achieves better adaptability and accuracy than the compared algorithms.

On the basis of the aforementioned cascading–parallel optimization, the programming tools Python 3.7.3 and Apache Spark 2.4 are used to develop parallel optimization algorithms for detecting rice diseases and predicting rice yield. In experimental section, the performance and accuracy of CPRF are evaluated according to the data collected in past 20 years.

*1.4. Organizations of the Paper*

The rest of the paper is organized as follows. Section 2 reviews the background and related work. Materials and methods are shown in Section 3. Section 4 gives PRF algorithm optimization in two aspects. Experimental methods, results and evaluations are shown in Section 5 with respect to the efficiency and accuracy. Finally, Section 6 gives the conclusion and discusses the future work.

## 2. Background and Related Work

*2.1. Background of Rice Yield Prediction*

Rice yield prediction is critical for early warning of food insecurity, agricultural supply chain management, and economic market [7]. Thus, the research on rice yield prediction is rather important for ensuring food security. Rice yield depends on interaction between temperature [8], precipitation [9], and plant diseases [10] as a continuum system [11]. Hence, they are much important inputs for the rice yield prediction system.

Numerous factors such as precipitation and temperature that affect rice yield, and effective monitoring of pests and diseases can improve rice yield in many provinces [12]. If disease monitoring and yield forecasting of rice is done well, it can bring about an increase in rice yield by targeting and reducing pest and disease infestation along with the season. The import and export trade of rice has a positive impact on reducing hunger among people in poor areas.

On the basis of the CPRF algorithm, a new sequence-based rice yield predictor, named CPRF-RY, is implemented. It regards the features of position-specific scoring matrixes, weighted average temperature of the month, and predicted relative rainfall as model inputs. Our results from a rice yield dataset and 17 validation datasets demonstrate that CPRF-RY outperforms any other predictors and can be applied as the main predictor in the agricultural field.

*2.2. Related Work*

There are three main models for rice yield prediction: (1) satellite-based models, (2) statistical machine learning models, and (3) random forest-based models. These methods have their strengths and weaknesses. Satellite-based models denote AquaCrop processing system for rice using crop monitoring [13], two types of vegetation related to crop yields were collected by using satellite [14], then image processing algorithms were implemented for predicting yield [15]. One study [16] used machine learning to process digital cameras images of the crops, the research [17] used mathematical functions to present a rice yield model. The paper [18] used a statistical machine learning model to predict

both global warming and corn production, and the study [19,20] forecast the crop yield with improvement by using machine learning techniques. The paper [21,22] enriched the crop yield prediction by using statistical machine learning models. Although the former algorithms have got relatively reasonable prediction results, the yield prediction accuracy is not satisfied in rice field accompanied with many rice leaf diseases [23]. Because these models depend on empirical knowledge and input parameters, they have difficulty in adapting to high dimensional data of different regions [24]. Statistical machine learning models, e.g., regression models and support vector machine regression algorithm [25], construct models based on the data and use the models to predict crop yield [26]. Generally, these models need to identify the large-scale and imbalanced features that have a significant influence on rice yield, which increases the complexity of the prediction task. The random forest-based model is inspired by efficiently running on large datasets and is a kind of artificial intelligence to solve feature expressions [27]. In the paper [28], although the authors investigated the effectiveness of random forest for crop yield prediction, they just used a simple data-intensive spatial interpolation to execute the prediction task and ignored the rice leaf disease features. Moreover, compared with traditional satellite-based models and statistical machine learning models, the cascade random forest method [29] is able to address the imbalanced data, resolves the negative effect of data imbalance by connecting multiple random forests in a cascade-like manner, and parallel random forest [30] perform a dimension-reduction approach in the training process and a weighted voting approach in the prediction process.

Former data processing algorithms have obtained relatively acceptable performance for low-dimensional or balanced small-scale datasets [24]. However, when imbalanced or large-scale data are encountered, these algorithms are always defective [30].

Given that agricultural plants are frequently attacked by several kinds of pests and diseases [10], data of agricultural diseases, such as rice planthopper, rice leaf roller, bacterial leaf blight, brown spot, and leaf smut are large-scale and imbalanced. Under these circumstances, processing these data with traditional algorithms is rather difficult [31]. For example, regarding the rice planthopper and rice leaf roller in southern provinces' farmlands, temperature, precipitation, rice planthopper, rice leaf roller, fertilizer, planting area of agricultural products, and affected area of agricultural products are seven important data sources. For a long time, the researchers from local institutes have tried traditional algorithms, and put them into practice for several years [32], but these were relatively unsatisfactory. Given that southern provinces' rice dataset is extremely complex, accompanied by the features of cross-regional character, high dimensionality [33], and large volume, the performance and accuracy of former data processing algorithms are significantly unsatisfactory.

Through the use of our method and algorithm, the recent result has proved that they can help in the large-scale, comprehensive wholesale market of agricultural products, as well as in the big data analysis [28] of the extent of temperature, precipitation, and product yield.

Determining how to decrease the plant diseases and insect pests in the entire country is a difficult problem because they could be affected by many factors, including climate changes, the number of beneficial birds, temperature, air, nutrients, soil condition, and rainfall and sunshine amount [34]. These influencing factors are collected from data sources across provinces, forming a massive agricultural plant dataset.

The prediction of plant diseases [10] can be divided into prevalence prediction, occurrence period prediction, and loss prediction in accordance with different prediction content and forecast quantity. The prediction time limit is more than 10 years, called long-term prediction.

The factors for predicting the prevalence of plant diseases should be selected from the host plant, pathogen, and environmental factors in accordance with the epidemic law of the disease [35]. In addition, the cultivation conditions, the number of mediators, and the growth and fertility status of the host plants ought to be considered [36].

## 3. Materials and Methods

### 3.1. Problem Formulation

According to the above-mentioned actual problem, symbolic variables are defined below, including input and output variables.

The initial training dataset is composed as $Z = \{(p_i, t_j), i = 1; 2; \ldots ; P; j = 1; 2; \ldots ; J\}$, where $p_i$ represents specific samples and $t_j$ represents the feature variable of dataset Z. In conclusion, the training dataset initially contains P samples, and J feature parameters exist in every sample, and the output variables are the accuracy of CPRFs.

### 3.2. Evaluation Indexes and Validation Procedure

Most experts have used six kinds of evaluation parameters, i.e., precision, recall, specificity, rice yield correlation coefficient (RYCC), $F\_\beta$ measure, and accuracy [29]. Likewise, these parameters were utilized here to evaluate the performance of predictive models, as depicted in the equations below. RYCC is a correlation coefficient between the observed rice yield and predicted classes of samples, and $F\_\beta$ measure is the weighted harmonic mean of recall and precision [26]. True positives (TPs) are correctly predicted increasing yield of rice and quantities of insects and pests, and true negatives (TNs) are correctly predicted decreasing yield of rice and quantities of pests and insects. False positives (FPs) refer to the number of decreasing yields of rice that were falsely predicted as increasing.

The above six indexes are dependent. $F\_\beta$ and RYCC determine a prediction model's overall performance.

### 3.3. CPRF

CPRF means an ensemble and high-speed method [27] that combines parallel optimization and independent current cascade trained decision trees. When precipitation and temperature are suitable for rice growth, various pest and disease infestations may disrupt the normal growth of rice, which in turn may affect the rice harvest.

Using CPRF, we can perform dimensionality reduction and de-imbalance operations with little impact on the accuracy of the analysis, and use historical data to analyze and predict the yield of rice, which in turn can guide the agricultural sector to formulate policies. Besides, CPRF algorithm integrates the advantages of a hybrid approach combining data parallel and task-parallel optimization by connecting multiple random forests in a cascade-like manner.

CPRF's architecture is made up of four parts: original training, random under sampling, balanced training, and obtaining the majority and minority classes. Its architecture is shown in Figure 1.

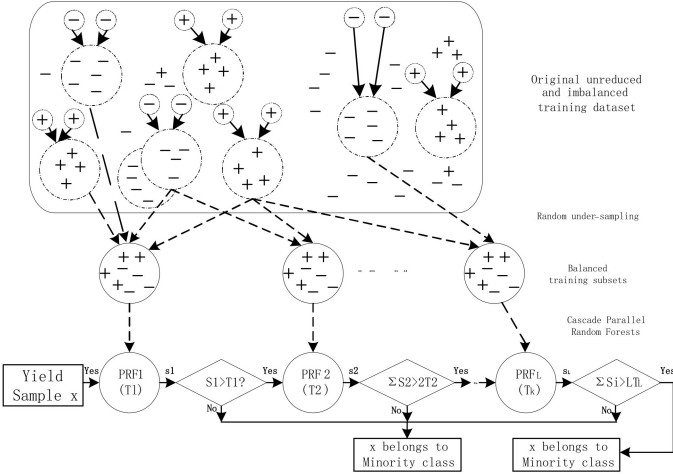

**Figure 1.** Architecture of CPRF.

The application of CPRF consists of three stages: small random forest convergence into integration forest, a prediction stage, and a training stage. In Figure 1, all dotted arrows represent the workflows in the later training stage, and the solid arrows represent the workflows in the prediction stage.

## 4. CPRF Algorithm Optimization

With the continual improvement in agricultural prediction accuracy for complicated and large-volume data, an optimization method for the CPRF algorithm is proposed. First, imbalance and dimension reduction approaches [20] are performed in the training process. Second, random under sampling and balanced training are implemented. Then, several minority classes are obtained. For example, pest's activity scope is a kind of minority class, which affects the harvest of agricultural plants. After regression and classification optimizations, the prediction accuracy of the algorithm is evidently improved.

The CPRF algorithm is a hybrid ensemble random forest algorithm and a former decision tree model. Original agricultural datasets are generated from j different training data subsets by utilizing a bootstrap sampling approach, and j decision trees should be built parallelly after the above subsets are trained [21].

*Cascade Random Forest Algorithm*

The construction steps of the CPRF algorithm are as follows.
Step 0. Preparation and initialization
For initialization, parameters CPRF←Ø; l←1 are set.
An empty string array $Z_j$ is created.
Step 1. Deciding whether to train CPRF and sampling j training subsets
j training subsets are sampled and classified, then whether to train CPRF is decided. From the original training dataset Z, the algorithm samples out j training subsets in a bootstrap sampling manner.
Step 2. Cascading each random forest and starting training
When the PRF is suitable for cascading [21], the training process can be started. In every node's splitting process, the gain ratio of each feature parameter is computed, and the optimal one is chosen as the splitting node.
Step 3. Converging k trees into the PRF model, then calculating entropy($Z_j$) for the target feature variable
The *k*-th trained trees are formed into a PRF algorithm defined as

$$H(P, \theta j) = \sum_{i=1}^{k} \mathrm{hi}(p, \theta j); \ (i = 1; 2; \ldots; k), \tag{1}$$

where $hi(p; \theta_j)$ represents a classical decision tree classifier, $P$ stands for the input feature vectors of the training dataset, and $\theta_j$ stands for an identically and independently distributed random vector [23] that determines the growth direction and the process.
Step 4. Returning the accuracy of CPRFs
The algorithm returns the accuracy value of CPRF, which we can compare with other algorithms.
The definition of the target variable's entropy in the training subset Sa(a = 1, 2, . . . , cl) is shown below.

$$\mathrm{Entropy}(\mathrm{Sa}) = \sum_{b=1}^{\mathrm{dv}} -\mathrm{p}_b \log \ \mathrm{p}_b, \tag{2}$$

In the upper equation, $p_b$ means b type value's probability, and dv means every value of variable in Sa.

$$\mathrm{Entropy}(Z_{\mathrm{ab}}) = \frac{1}{n} \sum_{a=1}^{n} \sum_{v \in V(Z_{ab})} \frac{\left| S_{(v,a)} \right|}{|S_a|} \mathrm{Entropy}(v(z_{ab})), \tag{3}$$

$Z_{\mathrm{ab}}$ means the b-th feature variable of Sab, b is a positive integer, and $V(Z_{\mathrm{ab}})$ is the set of feasible values of $Z_{\mathrm{ab}}$. $|S(v,a)|$ means the number of selected sample subsets $S(v,a)$.

Then, the information of self-split $I(Z_{ab})$ is calculated as follows.

$$I(Z_{ab}) = \frac{1}{n} \sum_{a=1}^{n} \sum_{b=1}^{d2} (-p(b,j) \log_2^{p(b,j)}), \tag{4}$$

In the above formula, $d_2$ means the quantity of different values of $Z_{ab}$, and $p(b,j)$ means the probability of the type of value b within all variables $Z_b$. We can generate the second averaging result through dividing by quantity n and then using entropy($S_a$) to divide the averaging result.

$$G(Z_{ab}) = \text{Entropy}(S_a) - \frac{1}{n} \sum_{a=1}^{n} \sum_{v \in V(Z_{ab})} \frac{|S_{(v,a)}|}{|S_a|} \text{Entropy}(v(z_{ab})), \tag{5}$$

As most research has depicted, the gain ratio of the feature parameter is shown as follows.

$$\text{GR}(Z_{ab}) = \frac{G(Z_{ab})}{I(Z_{ab})}, \tag{6}$$

Given that the dimension of the training dataset is reduced to an adaptive scale, the importance of each feature is computed in accordance with the parameters' gain ratio value. The importance of feature parameter $Z_{ab}$ is shown as follows.

$$\text{VI}(Z_{ab}) = \frac{1}{n} \sum_{a=1}^{n} \frac{GR(Z_{ab})}{\sum_{b=1} GR(Z_{ab})}, \tag{7}$$

The detailed training processing procedures are shown in Algorithm 1.

---

**Algorithm 1:** Balanced dimension reduction in the cascading process

---

$S_{maj}$—The majority class sample set; $S_{min}$—The minority class sample set;
Input: z* stands for imbalance coefficient;
TPR*: Using the threshold of each trained PRF as the true positive rate;
iTree: the number of trees for growing;
minNode: represents the minimum node size to split.
Output: $\text{CPRF} = \{(\text{PRF}_1, T_1), (\text{PRF}_2, T_2), \ldots, (\text{PRF}_j, T_j)\}$
Steps:
Step 0: Initialization, assigned to initial value
CPRF←Ø; l←1;
Step 1: Create an empty string array $Z_j$
Step 2: Decide whether the training array is a CPRF.
IF $|Z_{maj}| < = z^* \times |Z_{min}|$
Train a PRF, denoted as PRF1, on $Z_{maj} \cup Z_{min}$ with prescribed training parameters iTree and
$|\text{minNode}|$; assign an initial threshold T1; go to Step 10
END IF
Step 3: Train CPRFs.
While TRUE IF $|Z_{maj}| > z^* \times |Z_{min}|$, randomly sample a subset $Z_l$ from $Z_{maj}$ such that
$|Z_l| = |Z_{min}|$
else if $|Z_{maj}| > (1/z^*) \times |Z_{min}|$
then set $|Z_l| \leftarrow |Z_{maj}|$
else Go to Step 2
end if

---

Expand the formula
Step 4: Train a PRF on the balanced training subset Fl∪Fmin with parameters iTree and minNode.

The decision function of the random forest classifier at *n*-th layer is as follows:

$$H_n(x) = \text{sgn}(\frac{1}{n} \sum_{i=1}^{n} PRF_i(x) - T_n);$$

Step 5: Construct Tn such that the TP rate of predictions of $Hn(x)$ on $Z_{min}$
if all of the majority samples are zoomed out, which can be correctly predicted from
the current arrays $Z_{maj}$
then return $CPRF \leftarrow CPRF \cup \{(PRF_n, T_n)\}$
$n \leftarrow n + 1$
return $CPRF = \{(PRF_1, T_1), (PRF_2, T_2), \ldots, (PRF_j, T_j)\}$
end if
Step 6: After the upper training program, the classification accuracy of the algorithm
is described as follows.

$$CA_b = \frac{1}{n} \sum_{a=1}^{n} \frac{I(h_a(x) = z)}{I(h_a(x) = z) + \sum I(h_a(x) = w)} , \tag{8}$$

In the upper derivation, z means the value of correct class, and *w* means that is a value
of the error class. (*w*! = *z*).

After preliminary CPRF datasets are generated, the next step is training the agricultural
data. The detailed procedures are shown in Algorithm 2.

---

**Algorithm 2:** Using the CPRF algorithm to train the agricultural data in a cascading way

---

Input: $D_j$: the jth training dataset;
    Tyield: the yield table of CPRF;
    k: the number of important factors or variables selected by VI;
    m: the number of the selected feature variables.
Output: $Z_j$: a set of m important feature variables of $D_j$.
    $CPRF_{trained}$: the trained CPRF model.
    CPRF algorithm's Accuracy—The accuracy of the ensemble cascade parallel random forests
algorithm
Steps:
Step 1: for each feature variable $t_{ij}$ in $D_j$ do
Calculate Entropy ($t_{ij}$) for each input feature factor
Calculate gain $G(t_{ij}) \leftarrow Entropy(D_j) - Entropy(t_{ij})$;
Calculate split information
$I(t_{ij}) \leftarrow \sum_{a=1}^{c2} -p(a, j) \log_2(p_{(a,j)})$
obtain the gain ratio $GR(t_{ij}) \leftarrow G(t_{ij})/I(t_{ij})$
end for
Step 2: Arithmetic and get the value of variable importance
$VI(t_{ij}) \leftarrow \dfrac{GR(t_{ij})}{\sum_{a=1}^{M} GR(t_{(i,a)})}$ for feature variable $t_{ij}$
Step 3: Sort M feature variables in descending order by $VI(t_{ij})$
put top n feature variables to $F_j [0; \ldots ; n-1]$
define $c \leftarrow 0$;
for j = n to M − 1 do
    While c < (M − n) do
select $t_{ij}$ from (M − n) randomly;
    put $t_{ij}$ to $F_j[n + c]$;
    $c \leftarrow c+1$;
end while
end for
return $F_j$ Return $CPRF_{trained}$
Step 4: Determine the accuracy of the CPRF algorithm in comparison with other algorithms.

---

Compared with the traditional RF algorithm, this concatenating and parallel dimension reduction algorithm guarantees that the J selected feature variables are optimal when maintaining the same arithmetic complexity as the former algorithm.

## 5. Experiments Results and Analysis

In this section, the experimental analysis and accuracy of the CPRF algorithm are presented. We first show the selected datasets and discuss the experimental design. Next, on the basis of several factors that influence the yield of agricultural products, the original agricultural data are obtained from real-world farmland, and several algorithms are used. Meanwhile, we compare CPRF with other methods. Lastly, the classification accuracy of different algorithms is discussed. From the accuracy, we conclude that CPRF is the optimal algorithm.

Plant diseases and insect pests have a great impact on crop yields, and a land severely affected by diseases often faces no harvest or a small harvest. Therefore, we first conduct research on three common diseases of rice. Through the collection and analysis of data, such as pictures of diseased leaves, four random forest algorithms are used to predict, analyze, and compare the affected area data. Graphs are utilized to present the results.

### 5.1. Selected Datasets

To verify the proposed algorithm and compare it with a previous decision tree predictor, datasets collected by agricultural researchers from every year's official statistician were used as basic datasets. Eighteen datasets were collected. Among these datasets, the first 17 were utilized as test or training datasets, and the last one was utilized as an independent validation dataset.

The independent validation dataset was collected and sorted by a local agricultural department, namely, HuNan Province in China.

To confirm the accuracy and performance of the CPRF algorithm, two aspects of validation dataset were used. The first one mainly documented the temperature of rice fields in more than 15 years, and the second one mainly recorded the precipitation and insects that have affected rice yield for more than 15 years.

Table 1 lists the statistics of the agricultural datasets.

**Table 1.** Compositions of training datasets and aspects of the validation dataset.

| Training Dataset | | | Agricultural Validation Dataset | | |
|---|---|---|---|---|---|
| **Name** | **No. of Diseases** | **(numMin, numMaj)** | **Name** | **No. of Sequences** | **(numMin, numMaj)** |
| Train-dxs | 19,550 | (5619, 30,709) | Ytest95 | 95 | (1938, 16,319) |
| | | | RYtestset219 | 219 | (6098, 21,996) |

numMin is number of minority samples (pest and insect, manual intervention); numMaj is the number of majority samples (temperature and precipitation).

### 5.2. Precipitation and Temperature

Precipitation and temperature are two important aspects affecting the yield of rice. In dealing with these indicators that affect plant growth, the correlation between rice yield with precipitation and temperature can be expressed by the following equation.

$$Y(i,j) = \sqrt{\frac{\Gamma}{2}\left((1 - e^{-\frac{-|x_1(i)|}{\beta}})^{\alpha} + \left(1 - e^{-\frac{-|m*x_2(j)-k|}{\beta}}\right)^{\alpha}\right)}, \tag{9}$$

The precipitation and temperature values of the town in nearly 15 years were comprehensively considered. These data were collected and stored by local agricultural researchers yearly. The relationship between precipitation and temperature is depicted in Figure 2, where $X_1$ denotes the data of observed precipitation, $X_2$ denotes the data of observed temperature, and the z-axis represents the rice yield.

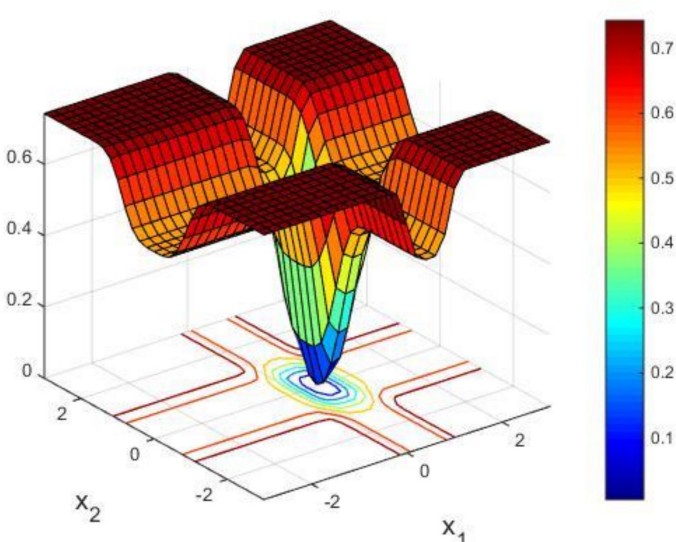

**Figure 2.** Precipitation and temperature relations.

In the above formulas, $X_1$ and $X_2$ represent the key factors that affect the yield of agricultural plants. Parameters m and k are variables that will change constantly in accordance with local conditions.

In Figure 2, the zero position of $X_1$-axis and $X_2$-axis represent extreme weather conditions, such as extreme drought and freezing conditions. Under such environmental conditions, the rice yield is close to zero, although the probability that the rice yield equals zero rarely occurs. The positive value of the $X_1$-axis represents the rainfall in spring and summer, and the negative value represents the rainfall in autumn and winter in one of the county-level observations. The positive and negative values of the $X_2$-axis approximately represent the cycle of rice lifespans. The surface soil temperature of different county-level observation points usually increases or decrease within the growth period. Suppose the theoretical maximum yield of rice is regarded as 100%, in the season with good weather conditions [12], due to the influence of plant diseases and insect pests, the maximum obtainable harvest only accounts for approximately 70%. Therefore, to increase the rice yield, the impact of various diseases and insect pests on plant growth should be studied and predicted.

### 5.3. Experimental Design of Rice Diseases

CPRF algorithm variants by using different parameters have been computationally studied. The algorithm follows the general yield of modern agriculture discussed above.

The collected data of relevant rice diseases and insect pests for at least 20 years are from a southern province of China. Three kinds of common rice diseases and insect pests are regarded as examples. Figure 3a–c are pictures of the three kinds of diseases, which are bacterial leaf blight, brown spot, and leaf smut, respectively. A good rice harvest is difficult to achieve if the leaves are affected by these diseases.

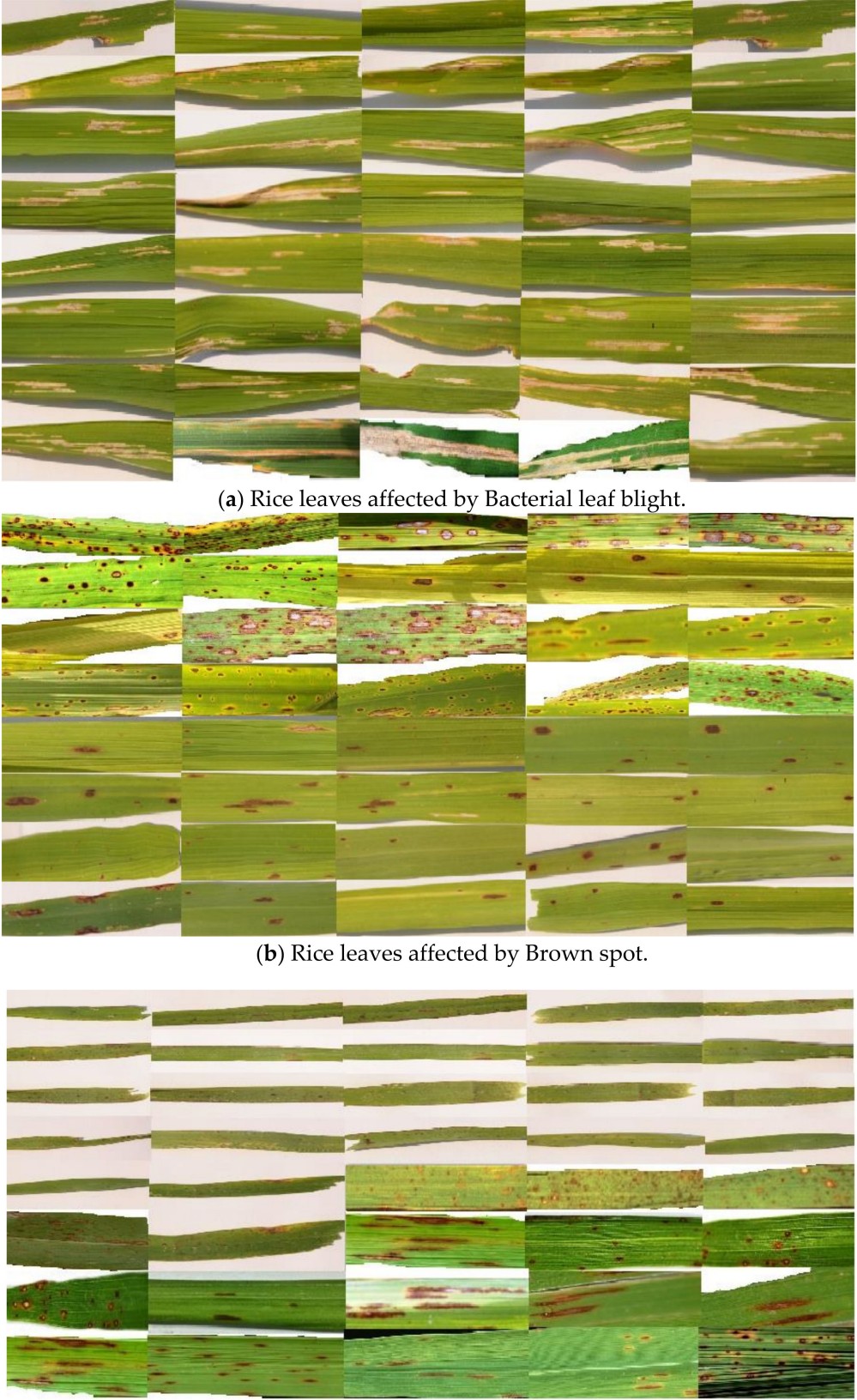

(**a**) Rice leaves affected by Bacterial leaf blight.

(**b**) Rice leaves affected by Brown spot.

(**c**) Rice leaves affected by Leaf smut.

**Figure 3.** (**a**) Rice leaves affected by bacterial leaf blight; (**b**) Brown spot; (**c**) Leaf smut.

The pictorial data above and other numerical data are integrated into structured and unstructured databases.

### 5.4. Advantages of CPRF Algorithm

The data of the rice field for the recent 20 years are trained with four kinds of algorithms, namely, CPRF, CRF, Spark-MLRF, and random forest.

In CPRF, data communication operations exist in the process of allocating data and training process. From the entire parallel training process of CPRF in the Spark cluster, this cascade parallel optimization approach achieves a larger storage and better workload balance than former algorithms.

Combining the abovementioned data of several historical rice diseases, at least four algorithms are used to analyze and then compare with one another, aiming to find the best algorithm. The comparison result is shown in Figure 4.

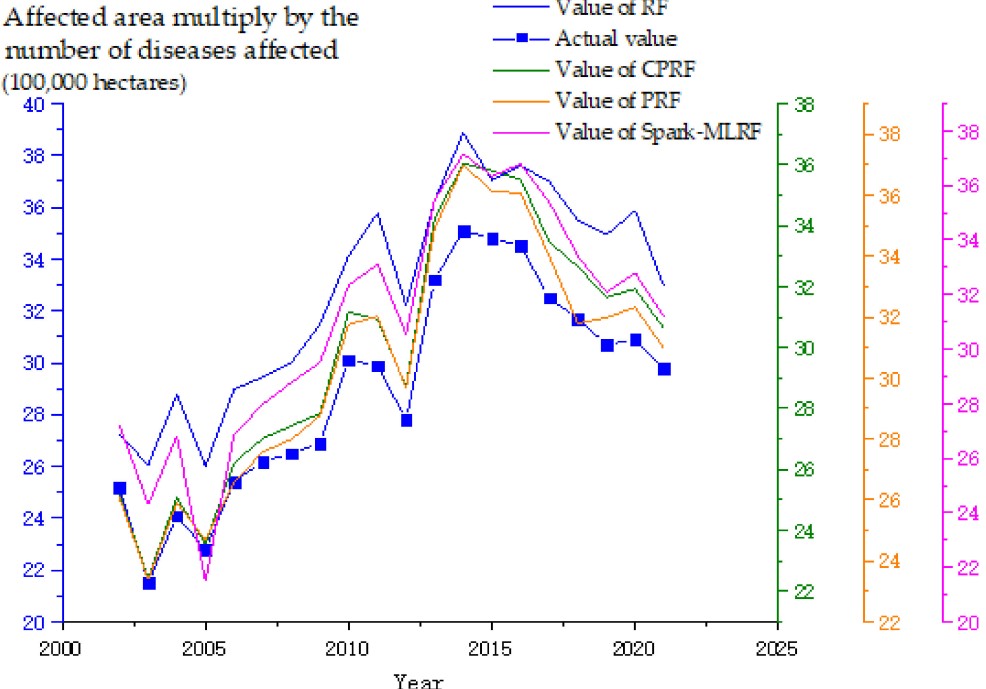

**Figure 4.** Comparison of four algorithms by using data of an affected area in 20 years.

The planting area of the province has been approximately 14 million to 16 million hectares in the recent 20 years. Apart from policy influence and natural disasters, the farmland area has grown slowly in the past 20 years. The algorithm is used to analyze the planting area and compare the result with the actual data. The comparison of 20 sets is shown in Figure 5.

In the rice cultivation aspect, the percentage of the area affected by several diseases to the entire area of farmland has been less than 3% in recent years. However, every percent multiplied by the total rice yield is a large number, which also can save many people from hunger. A comparison of the actual rate with the four algorithms' prediction rates is shown in Figure 6. The result indicates that the CPRF algorithm's prediction is the closest to the actual rate.

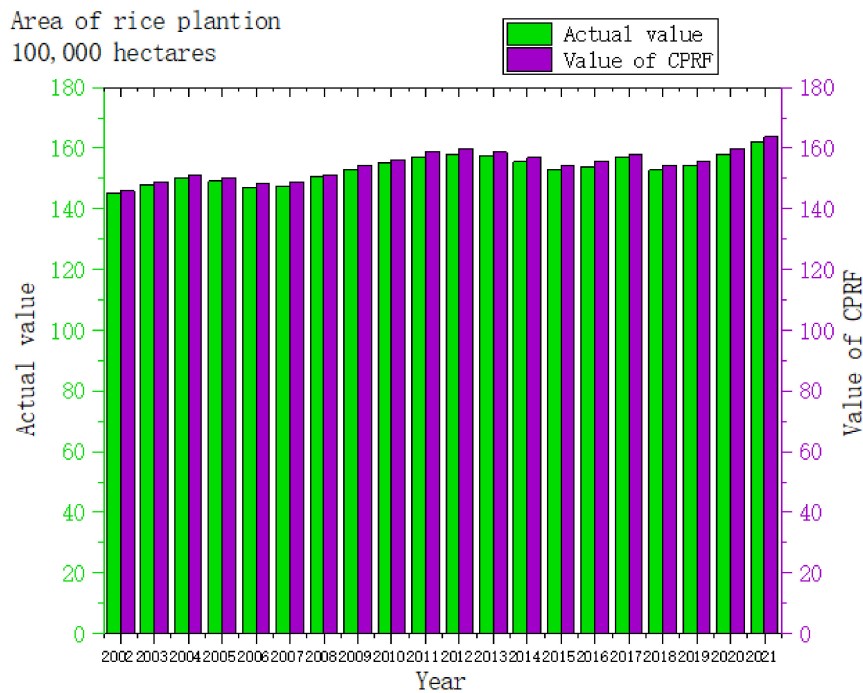

**Figure 5.** Planting area comparison of actual value and CPRF prediction in 20 years.

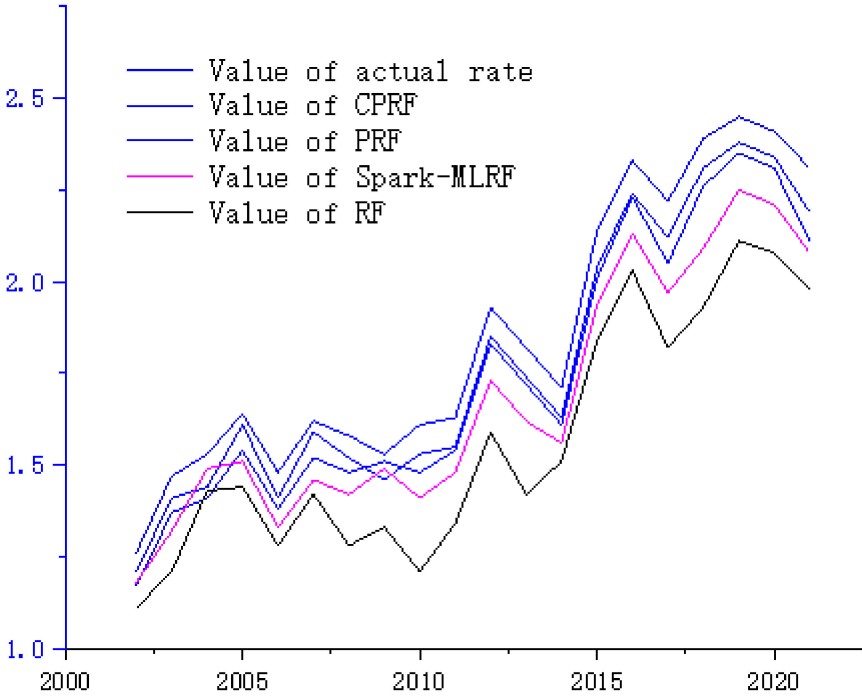

**Figure 6.** Comparison of four algorithms by using the proportion of affected area.

### 5.5. Efficiency Comparison for Different Algorithms

The algorithms' efficiency is very important, because the data of rice or wheat crops is huge, and the farmers want the proper advice in a short time, they must work for a long time in farmland and they usually spend little patience on waiting the computing

results, so the efficiency is quite important. CPRF algorithm has several features as follows 1. Reliability and high fault tolerance. The system crash on one server will not affect other servers. In the CPRF distributed computing system, more machines can be added if needed. 2. Scalability, In the CPRF distributed computing system, more machines can be added as needed. 3. Flexibility. CPRF system can easily install, implement, and debug new services. 4. Fast calculation speed. CPRF distributed computer system can have the computing power of multiple computers, making it faster than other systems. 5. Openness. Since CPRF is an open system, the service can be accessed locally and remotely. 6. High performance, compared with a centralized computer network cluster, CPRF system can provide higher performance and better performance.

The parameter "n_job" refers to the number of processes that can be used. When n_job equals two, the prediction time, which can also be named train_time, is 44.76 s. As n_job increases, the train_time of other algorithms may decrease at first and then increase, as shown in Figure 7.

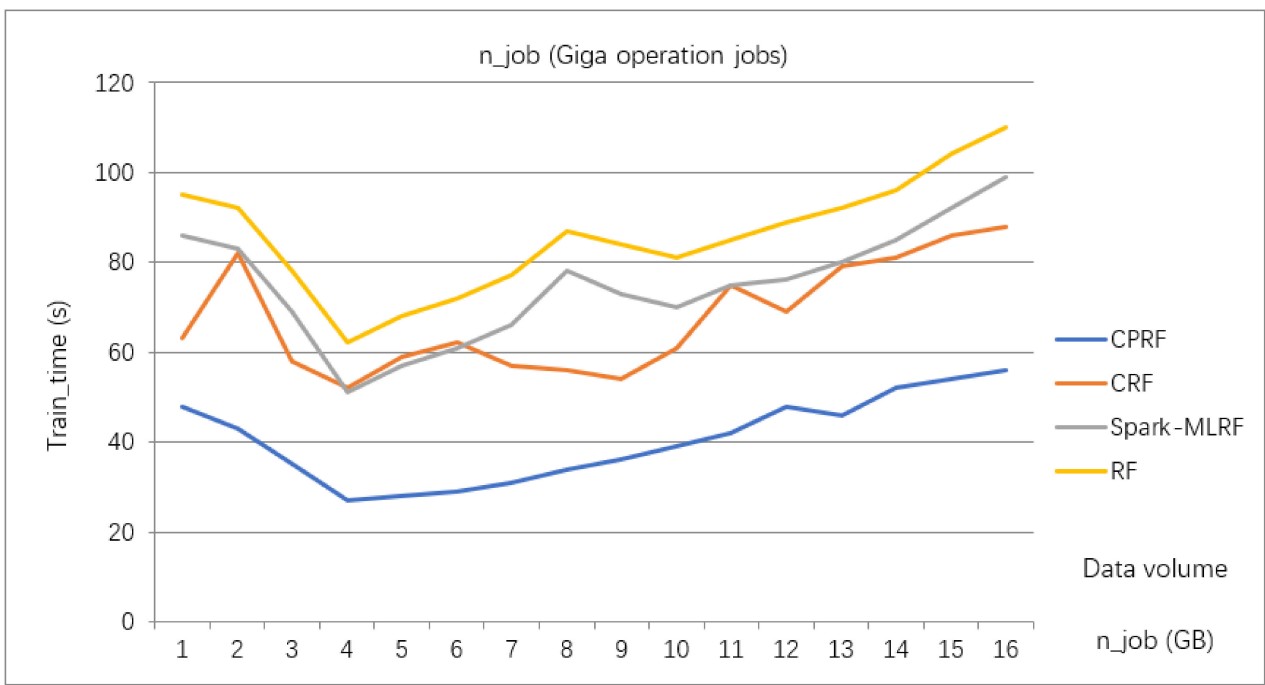

**Figure 7.** Train_time along with the job quantity of CPRF and other algorithms.

The CPRF algorithm copes with the same agricultural condition. The train_time of the prediction process decreases as n_job increases. However, when n_job is more than 12, train_time is basically consistent, as demonstrated in Figure 7. From the above experiment, the advantages of the CPRF algorithm can be clearly depicted.

*5.6. Average Train_Time Comparison for Different Algorithms*

To verify the accuracy of CPRF, several experiments are performed for the random forest, CRF, Spark-MLRF, and CPRF algorithms. With the different data volumes and qualitative running times, Figure 8 elucidates the comparison of the four algorithms.

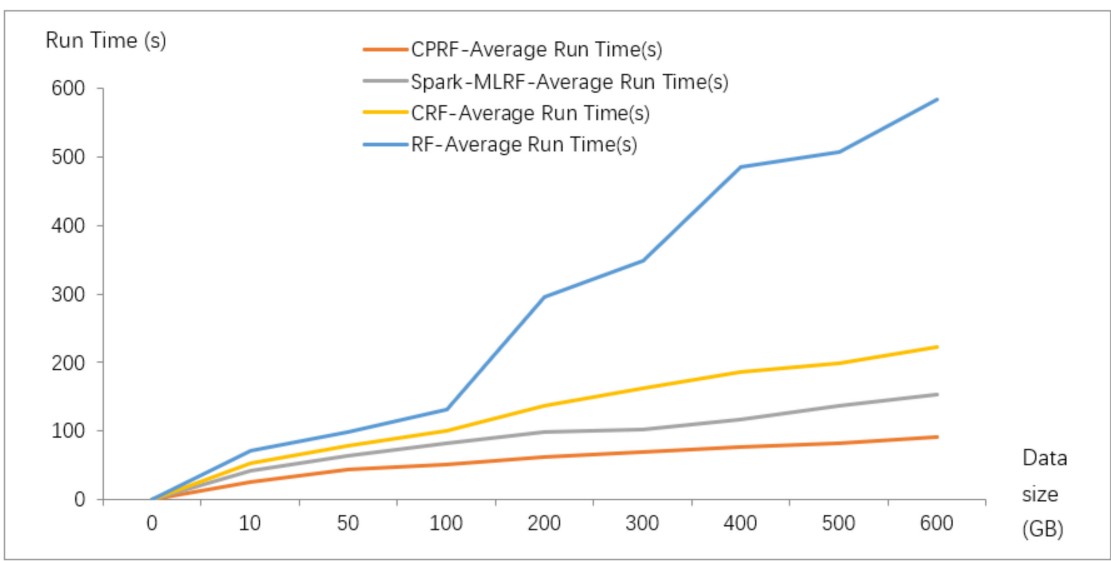

**Figure 8.** Average running time of the different algorithms.

Agricultural big data server clusters in a certain area are used for calculations [14], and the pros and cons of the algorithms can be judged by average running time and accuracy.

As depicted in Figure 8, by using the agricultural datasets for numerous experiments, which are performed to compare the time consumption of CPRF with those of random forest, CRF, and Spark-MLRF, the results show that the CPRF algorithm uses the least average running time.

In traditional classification learning methods, classification accuracy is often used as an evaluation index, but this index is unsuitable for unbalanced datasets. The unbalanced dataset uses F_$\beta$ measure and RYCC as evaluation indicators. These two indicators come from the confusion matrix, as shown in Table 2. As explained in Section 5.2, TP represents the number of positive samples, TN represents the number of negative samples, and FN and FP represent the samples whose judgment errors are positive and negative, respectively.

**Table 2.** Confusion matrix.

| Classification | Predict Positives | Predict Negatives |
|---|---|---|
| Actual positives | True Positives (TP) | False Negatives (FN) |
| Actual negatives | False Positives (FP) | True Negatives (TN) |

F_$\beta$measure is a classification evaluation index that comprehensively considers recall and precision, and (+) means that the larger, the better:

$$\text{F\_}\beta \text{ measure} = \frac{n(1 + \beta^2) * recall * precision}{\beta^2 * recall + precision}, \tag{10}$$

In the above formula, precision is the accuracy of the search, recall is the ratio of the total search, and $\beta$ takes the value $[0, \infty]$.

F_$\beta$ measure can be used to investigate the trade-off between recall and accuracy. When $\beta < 1$, the role of precision is emphasized; when $\beta > 1$, the role of recall is emphasized. The definition of precision and recall is as follows:

$$precision = (TP)/(TP + TP), recall = TP/(TP + FN).$$

The RYCC value represents the geometric mean of the classification accuracy of the positive class and the classification accuracy of the negative class. (+) means that bigger is better, and its definition is as follows:

$$\text{RYCC}(+) = \sqrt{\frac{TP}{TP + FP} * \frac{TN}{TN + FP}} \ , \tag{11}$$

The metric RYCC is employed to evaluate the classification accuracy according to the observed classes and predicted classes in test dataset, while maintaining the classification accuracies of the negative and positive classes. That is, the value of RYCC is the largest only when the classification accuracies of the negative and positive classes are both high. This study uses RYCC and $F_\beta$ measure to deal with the overall imbalanced classification performance of the dataset.

### 5.7. Average Execution Time of Different Datasets

More kinds of pests and diseases are acquired for different plant diseases and insect pests' datasets by using the CPRF algorithm in the case of different numbers of computing nodes to discover the pattern by using related experiments. The results show that the speedup data decrease with the increase in the number of slave nodes, as illustrated in Figure 9.

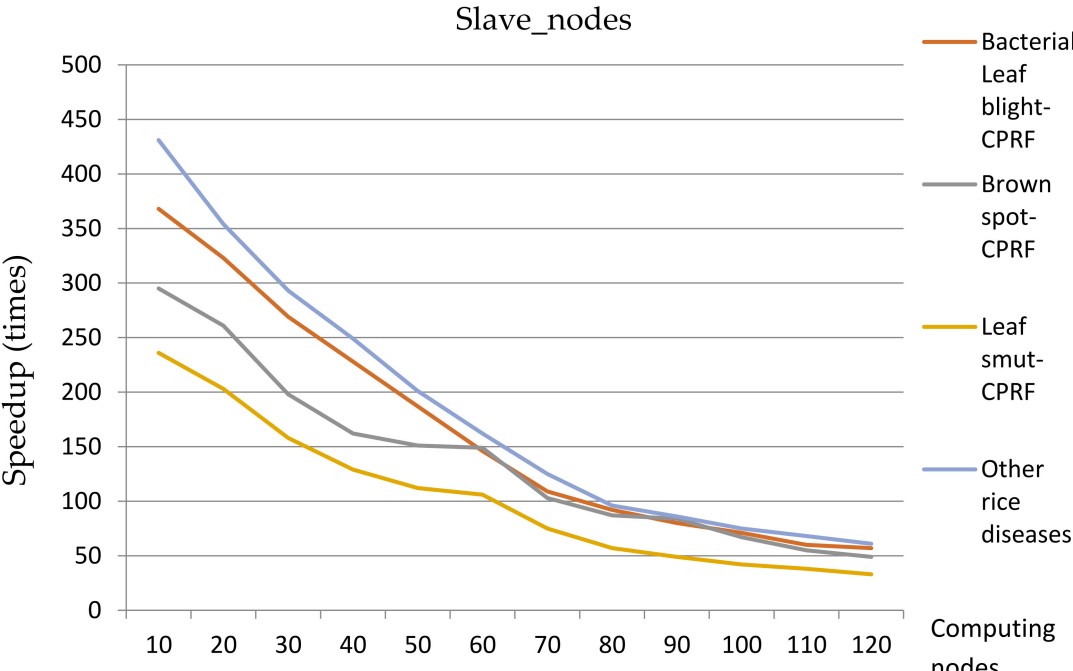

**Figure 9.** Average execution time of CPRF with different scales of agricultural plant diseases.

Figure 9 depicts the calculation results of the datasets of seven common plant diseases and insect pests, namely, bacterial leaf blight, brown spot, leaf smut, rice leaf roller, rice planthopper, and other plant diseases. Given the benefits of cluster environment and cascade parallel algorithm, the computational speed of CPRF tends to increase in the experiments. At the same time, with the increase in computing slave nodes (physical machine or virtual machine) in the server cluster, the average calculation time decreases slowly.

### 5.8. Accuracy Comparison of Different Algorithms

Most of the experiments are performed on Python and Spark environments, which are built of a master node and more than 200 slave nodes. Every node is executed in Centos 7.3 and has one Core (i7) Quarter-Core 2.50 GHz CPU and 32 GB memory.

Plant growth data in several regions are used. Under the same precipitation and temperature, the four algorithms are used to predict the occurrence of six different pests and diseases. The runtime of various algorithms is determined in the calculation process. The four prediction results are compared with actual data to determine the accuracy of the algorithms. We evaluate the computation accuracy of CPRF via comparison with random forest, CRF, and Spark-MLRF.

The accuracy value equals to prediction value of the algorithms dividing the actual value, the prediction accuracy of different algorithms is described as follows.

$$\text{Accuracy}_{\text{prediction}} = 1 - \left| 1 - \frac{\text{Value}_{(\text{trained CPRF})}}{\text{Value}_{(\text{actual yield})}} \right|, \tag{12}$$

The proportion of the correct prediction is the accuracy. Table 3 presents the accuracy comparison of these algorithms.

**Table 3.** Comparison of CPRF with other algorithms.

| The Algorithms | Accuracy Value |
|---|---|
| CPRF | 96.253% |
| CRF | 92.321% |
| Spark-MLRF | 86.159% |
| Random Forest | 75.072% |
| Non-Linear | 63.084% |

Table 3 demonstrates that the highest accuracy value among the different random forest algorithms, which is highlighted in bold face, comes from the CPRF algorithm. The second highest value is obtained by the PRF algorithm, which also uses distribution algorithms. The third one is generated by Spark-MLRF, which uses machine learning methods. The lowest value is from the RF algorithm, which exhibits the largest forecast error.

The paper also compares the prediction accuracy with that of the cited references, and the comparison results are shown in the following Table 4.

**Table 4.** Comparison of paper with cited literatures.

| The Algorithms | Accuracy Value | Cited Reference |
|---|---|---|
| CPRF | 96.253% | —- |
| Quad-Pol RadarSAT-2 and Random Forest | 95.94% | [37] |
| CNN Optimization | 94.27% | [38] |
| BORO Rice Yield Estimation | 94.2% | [39] |
| Rice Production using Improvised NDVI Threshold | 93.72% | [40] |

## 6. Conclusions

Predicting and increasing agricultural plant yield are difficult problems for many organizations nowadays. The use of big data technology and CPRF algorithm to analyze plant diseases and insect pests is relatively correct and suitable. This paper proposes a CPRF algorithm to analyze the agricultural yield problems.

We believe that this study provides suitable algorithms and tools to explore the enormous amount of plant influencing factors in complex networks. In this dissertation, the CPRF algorithm was implemented in an agricultural area with big data. The accuracy of the algorithm was improved by adopting key dimensions and parallel methods. A cascade

and weighted voting approach was constructed to find the least rice disease and highest yield in the agricultural field, which was effective. From the experiment, different results could be obtained by adjusting various parameters. The CPRF algorithm was more accurate than several other algorithms, which were utilized to solve the problems.

The limitation of this paper is that its model training process relies on offline data training, and the data needs to be collected and pre-processed manually. The original data collection may be optimized with semi-supervised learning. According to the present experiment, the method in this paper is better suited for the local application in agricultural field. With the CPRF technology optimized and increasing accumulation data, the CPRF algorithm based on big data is gradually formed.

**Author Contributions:** Conceptualization, L.Z. and L.X.; methodology, software, validation, L.Z. and L.X.; writing, original draft preparation, L.Z.; writing, review and editing, L.X. and C.H.; supervision, L.X.; project administration, Z.W. and C.H. All authors have read and agreed to the published version of the manuscript.

**Funding:** This research was funded by the National Key R&D Program of China, grant number. 2018YFC2001700. Beijing Natural Science Foundation grant number L192005.

**Conflicts of Interest:** The authors declare no conflict of interest.

## Abbreviations

The nomenclature and abbreviation list.

| Num | Abbreviation | Full Name |
| --- | --- | --- |
| 1 | RF | Random Forest |
| 2 | PRF | Parallel Random Forest |
| 3 | CPRF | Cascade Parallel Random Forest |
| 4 | MLRF | Machine Learning Random Forest |
| 5 | kNN | K-Nearest Neighbor |
| 6 | RYCC | Rice Yield Correlation Coefficient |
| 7 | CPRF-RY | Cascade Parallel Random Forest-Rice Yield |

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
