# Peer review of "Cascade Parallel Random Forest Algorithm for Predicting Rice Diseases in Big Data Analysis"

_electronics, doi:10.3390/electronics11071079_

Round 1

Reviewer 1 Report

Dear authors, while the presentation is nice in shape, there are a few comments and/or suggestions to improve the manuscript. While there is some interest in this type of research, I found this paper only mildly interesting in its present form, and accordingly not worthy of publication. Please strongly consider the following suggestions:

  1. The significance of the study is not clear to me and there is a serious literature review gap in this paper. I strongly consider that the distinguished authors did not pay accordingly attention to the relevant cited references. In this context, please clarify better the advantages of this paper in the introduction section because in the literature all cited recent papers consider the same proposed approach. I strongly recommend the authors reconsider the related work section for the literature review and discuss the drawbacks of existing works.
  2. I suggest that in the Introduction section, the value-added of this paper should be explained. How is it different from other papers in the field? What novelty does it offer? In this sense, in the Results Evaluation section, the authors must highlight the advantages of the proposed approach compared with the literature. Please emphasize the real contribution, considering the comments from 1.
  3. Please describe the necessity of each reference in the first three sections, because multiple citations contradict the ethics of journals with international visibility. Please also reconsider [1-3], [4-7],....[13-22], [32-37]...... etc.
  4. The authors must present the limitation of the considered method.
  5. The Experimental Results and Analysis section should be revised to improve the impact of the paper. The results are abundant but the results’ analysis is not enough. However, I cannot see deeply analysis related to them and cannot understand the meaning of results. Please add more analysis.

Minor revisions:

  1. What means MLRF; kNN ; RYCC ; PRF … etc. Please add a nomenclature and abbreviation lists
  2. The related work Section is with only four references (3, 64, 65, 66). Please explain.
  3. Between lines 167-175 you use different notation: (X1, X2), (x1, x2) or (x1, x2). Please use the only one.

However, the theoretic background is solid and the article is enhanced with very interesting results depicted in the last section. The conclusions section sums up the research output while the readers can find rich information for further study.

Author Response

Dear Reviewer,

We would like to thank the editor and both of the reviewers for providing very insightful comments that helped us improve the quality and completeness of this manuscript. We have carefully considered all your comments and reflected them to prepare this revision. In the following responses, we transcribe the reviewers’ comments and provide point-by-point responses.

In this response letter (word version in attachment), we label the textual responses via red font. The content in revised manuscript is labeled as black font or be shown as a screenshot. The key content is labeled via yellow background.

Kindly see below for detailed responses.

Sincerely Yours,

Authors

Point 1: The significance of the study is not clear to me and there is a serious literature review gap in this paper. I strongly consider that the distinguished authors did not pay accordingly attention to the relevant cited references. In this context, please clarify better the advantages of this paper in the introduction section because in the literature all cited recent papers consider the same proposed approach. I strongly recommend the authors reconsider the related work section for the literature review and discuss the drawbacks of existing works.

 Response 1: We thank the reviewer for the constructive comments.

Compared with existing methods, this paper is more suitable for the problem of prediction and classification of agricultural data with high-dimensional inputs and unbalanced data categories.

In the introduction section of revised version, we have added a description of the advantages of our approach.

The yield of agricultural products is considerably important for every country [1]. The Food and Agriculture Organization of the United Nations (FAO) predicted that the demand for agricultural production will be increased by 70 % to sustain the subsistence of 9 billion people in 2050 [2]. As one of the most important major food sources, rice sustains more than 50% of worldwide people and significantly contributed to global food security [3]. Therefore, rice yield has a great impact on the economy and politics. Food shortages hinder economic development and even lead to social unrest; thus, food security is very important for developing countries [4]. On the other hand, although developed countries have abundant agricultural product resources, the agricultural plant protection should also be taken seriously [5], including crops yield prediction and botany disease prevention. Therefore, it is significant to predict rice yield and diseases to maintain the crop yield and ensure food security.

The analysis of the agricultural plant diseases that influence rice yields has motivated the study of cascade parallel random algorithm based on big data. The search for agricultural botany protection has become one of the widest computational problems in the big data field. Plant diseases, such as rice planthopper, bacterial leaf blight, brown spot, leaf smut, and rice leaf roller, are adverse to crop growth. The extent of plant diseases and insect pests is related to the output of agricultural products, such that data on plant diseases and insect pests should be efficiently dealt with urgently.

In the Related Work section, we added the description of the drawbacks of existing works.

There are three main models for rice yield prediction: (1) satellite-based models, (2) statistical machine learning models, and (3) random forest-based models. These methods have their strengths and weaknesses. Satellite-based models denote AquaCrop processing system for rice using crop monitoring [13], two types of vegetation related to crop yields were collected by using satellite [14], then image processing algorithms were implemented for predicting yield [15]. One study [16] used machine learning to process digital cameras images of the crops, the research [17] used mathematical functions to present a rice yield model. The paper [18] used a statistical machine learning model to predict both global warming and corn production, and the study [19] and [20] forecast the crop yield with improvement by using machine learning techniques. The paper [21] and [22] enriched the crop yield prediction by using statistical machine learning models. Although the former algorithms have got relatively reasonable prediction results, the yield prediction accuracy is not satisfied in rice field accompanied with many rice leaf diseases [23]. Because these models depend on empirical knowledge and input parameters, they have difficulty in adapting to high dimensional data of different regions [24]. Statistical machine learning models, e.g., regression models and support vector machine regression algorithm [25], construct models based on the data and use the models to predict crop yield [26]. Generally, these models need to identify the large-scale and imbalanced features that have a significant influence on rice yield, which increases the complexity of the prediction task. The random forest-based model is inspired by efficiently running on large datasets and is a kind of artificial intelligence to solve feature expressions [27]. In the paper [28], although the authors investigated the effectiveness of random forest for crop yield prediction, they just used a simple data-intensive spatial interpolation to execute the prediction task and ignored the rice leaf disease features. Moreover, compared with traditional satellite-based models and statistical machine learning models, the cascade random forest method [29] is able to address the imbalanced data, resolves the negative effect of data imbalance by connecting multiple random forests in a cascade-like manner, and parallel random forest [30] perform a dimension-reduction approach in the training process and a weighted voting approach in the prediction process.

Point 2: I suggest that in the Introduction section, the value-added of this paper should be explained. How is it different from other papers in the field? What novelty does it offer? In this sense, in the Results Evaluation section, the authors must highlight the advantages of the proposed approach compared with the literature. Please emphasize the real contribution, considering the comments from 1.

Response 2: Many thanks for the reviewer’s comment.

In terms of the innovation of the paper, compared with the CRF , Spark-MLRF, RF method, the proposed CPRF algorithm uses a stacked cascade model which reduce the dimensionality of the data and improve the prediction efficiency and accuracy. In addition, this paper performs a downsampling operation on the original training set, which can solve the problem of data category imbalance.

In the Introduction section, the novelty and advantages of the method of this paper are introduced.

The proposed CPRF algorithm combines the advantages of task parallelism and data cascade. For high dimensional and imbalanced agricultural data, CPRF achieves better adaptability and accuracy than the compared algorithms. 

In the experimental section, as shown in Figure 4 to Figure 9 and Table 3 to Table 4, the proposed method in this paper outperforms the compared methods in terms of prediction accuracy and time consumption.

We excerpt major revisions as follows:

Point 3: Please describe the necessity of each reference in the first three sections, because multiple citations contradict the ethics of journals with international visibility. Please also reconsider [1-3], [4-7],....[13-22], [32-37]...... etc.

Response 3: Many thanks for the reviewer’s comment. We revised the citations contradict the ethics of journals with international visibility in the first three sections.

We reduced the number of references to 40, and placed the citations to the references as well as the introductions in Related Work. The modified reference citations are presented as follows.

There are three main models for rice yield prediction: (1) satellite-based models, (2) statistical machine learning models, and (3) random forest-based models. These methods have their strengths and weaknesses. Satellite-based models denote AquaCrop processing system for rice using crop monitoring [13], two types of vegetation related to crop yields were collected by using satellite [14], then image processing algorithms were Implemented for predicting yield [15]. One study [16] used machine learning to process digital cameras images of the crops, the research [17] used mathematical functions to present a rice yield model. The paper [18] used statistical machine learning model to predict both global warming and corn production, and the study [19] and [20] forecast the crop yield with improvement by using machine learning technique. The paper [21] and [22] enriched the crop yield prediction by using statistical machine learning models. Although the former algorithms have got relatively reasonable prediction results, the yield prediction accuracy is not satisfied in rice field accompanied with many rice leaf diseases [23]. Because these models depend on empirical knowledge and input parameters, they have difficulty in adapting to high dimensional data of different regions [24]. Statistical machine learning models, e.g., regression models and support vector machine regression algorithm [25], construct models based on the data and use the models to predict crop yield [26]. Generally, these models need to identify the large-scale and imbalanced features that have a significant influence on rice yield, which increases the complexity of the prediction task. A random forest-based model is inspired by efficiently running on large datasets and is a kind of artificial intelligence to solve feature expressions [27]. In the paper [28], although the authors investigated the effectiveness of random forest for crop yield prediction, they just used a simple data-intensive spatial interpolation to execute the prediction task and ignored the rice leaf disease features. Moreover, compared with traditional satellite-based models and statistical machine learning models, cascade random forest method [29] is able to address the imbalanced data, resolves the negative effect of data imbalance by connecting multiple random forests in a cascade-like manner, and parallel random forest [30] perform a dimension-reduction approach in the training process and a weighted voting approach in the prediction process.

Point 4: The authors must present the limitation of the considered method.

Response 4: Many thanks for the reviewer’s comment. We add the limitation of the considered method in the Section of Conclusion.

The limitation of this paper is that its model training process relies on offline data training, and the data needs to be collected and pre-processed manually.

In the conclusion section, we add a description of the limitations of the method in this paper and future research prospects to address the limitations.

The original data collection may be optimized with semi-supervised learning. According to the present experiment, the method in this paper is better suited for the local application in agricultural field.

We excerpt major revisions as follows:

Point 5: The Experimental Results and Analysis section should be revised to improve the impact of the paper. The results are abundant but the results’ analysis is not enough. However, I cannot see deeply analysis related to them and cannot understand the meaning of results. Please add more analysis.

Response 5: We thank the reviewer for the constructive comment.

Figure 10 from the original paper was removed because of the redundancy between the experimental results presented in this figure and the results in Table 3.

Besides, from subsection 5.4 to subsection 5.8 of Experiment results’ analysis, the method in this paper outperforms the three methods compared in terms of average train_time, average execution time, efficiency and accuracy of its algorithm execution. Because this cascade parallel optimization approach achieves a higher efficiency, less execution time and better workload balance than former algorithms, makes prediction accuracy of CPRF algorithm higher than other algorithms. Furthermore, the analysis of the experimental results compared with accuracy of the references has been added to the paper in Table 4.

 We excerpt major revisions as follows:

Minor revisions:

Point 6: What means MLRF; kNN ; RYCC ; PRF … etc. Please add a nomenclature and abbreviation lists.

Response 6: Many thanks for the reviewer’s comment. We added Table 5 belo4w in Appendix A.

Table 5. The nomenclature and abbreviation list.

Num

Abbreviation

Full Name

1

RF

Random Forest

2

PRF

Parallel Random Forest

3

CPRF

Cascade Parallel Random Forest

4

MLRF

Machine Learning Random Forest

5

kNN

K-Nearest Neighbor

6

RYCC

Rice Yield Correlation Coefficient

7

CPRF-RY

Cascade Parallel Random Forest-Rice Yield

Point 7: The related work Section is with only four references (3, 64, 65, 66). Please explain.

Response 7: Many thanks for the reviewer’s comment.

The introduction to the reference in the original Introduction is placed in Related Work section. Furthermore, we streamlined, optimized number of references and revised the content of the related work Section.

We excerpt major revisions as follows:

Point 8: Between lines 167-175 you use different notation: (X1, X2), (x1, x2) or (x1, x2). Please use the only one.

Response 8: Many thanks for point it out in the reviewer’s comment. Between former lines 167-175, we used different notation as (X1, X2), (x1, x2) or (x1, x2). We used the new notation as (X1, X2) in the paper. We excerpt major revisions as follows:

Reviewer 2 Report

Although authors performed several changes due to the first round of review, many other have still not been addressed:

In the Introduction section:

  • Lines 48-50: “agriculture techniques … are mostly unsuitable for agricultural plant protection given many kinds of influencing factors”. What is the arguments authors use to make this statement?

In section “2. Background and related work”

  • Line 92: “Third, these data factors affecting rice yield are high dimensional and large scale” still makes no sense.
  • Line 106: “Though former data transaction algorithms have obtained relatively acceptable performance for low dimensional or balanced small-scale datasets should be scientific supported with references.
  • Line 116-118: “For a long time, the researchers from local institutes tried traditional algorithms, and put them into practice for several years, but these were ineffective.” should be scientific supported with references.

In section “3. MATERIALS AND METHODS”

  • Lines 144-146: the authors say they use temperature dataset recorded for more than 15 years but they do not specify the period of time for “the second dataset mainly recorded the precipitation and insects which affects wheat yield.”
  • The subsection 3.2 “Evaluation Indexes and Validation Procedure” has no references although authors state that “Most experts…” and describe different parameters which they are defined in specialized literature.

In section “4. CPRF ALGORITHM OPTIMIZATION”

  • Subsections 4.1 title makes no sense “Initial of Cascade Random Forest Algorithm”.

In section “5. CPRF ALGORITHM OPTIMIZATION”

  • The authors failed to provide a formula for accuracy values in Table 2.
  • Line 464: “RYCC is to obtain the accuracy of the two categories..” should be rephrased.
  • Figure 9 presents several agricultural disease data like Rice planthopper, Rice Leaf Roller, Wheat Stripe Rust which are nowhere mentioned in the text previously.
  • The results in Figure 10 cannot be verified since there is no formula or explanation for the accuracy. So far, from everything authors presented, it is difficult to believe that their CPRF implementation leads to an almost 100% accuracy.
  • One main disadvantage of this work is that it is not comparing the results with the already existing ones in literature. Therefore, the presented results are questionable.

Author Response

Dear Reviewer,

We would like to thank the reviewer for providing very insightful comments that helped us improve the quality and completeness of this manuscript. We have carefully considered all your comments and reflected them to prepare this revision. In the following responses, we transcribe the reviewers’ comments and provide point-by-point responses.

In this response letter, we label the textual responses via red font. The content in revised manuscript is labeled as black font or be shown as a screenshot. The key content is labeled via yellow background.

Please download the response letter in this attachment.

Kindly see below for detailed responses.

Sincerely Yours,

Authors

Response to Reviewer  Comments

Point 1: In the Introduction section:

  • Lines 48-50: “agriculture techniques … are mostly unsuitable for agricultural plant protection given many kinds of influencing factors”. What is the arguments authors use to make this statement?

Response 1: Many thanks for the reviewer’s comment.

After careful research and analysis, we found that this statement was not rigorous, so we removed the sentence “agriculture techniques … are mostly unsuitable for agricultural plant protection given many kinds of influencing factors.” in the former Lines 48-50.

Point 2: In section “2. Background and related work”

  • Line 92: “Third, these data factors affecting rice yield are high dimensional and large scale” still makes no sense.
  • Line 106: “Though former data transaction algorithms have obtained relatively acceptable performance for low dimensional or balanced small-scale datasets should be scientifically supported with references.
  • Line 116-118: “For a long time, the researchers from local institutes tried traditional algorithms, and put them into practice for several years, but these were ineffective.” should be scientific supported with references.

Response 2: We thank the reviewer for the constructive comment.

(1) We also found that this statement was not rigorous, so we deleted the sentence in the former Line 92 “Third, these data factors affecting rice yield are high dimensional and large scale”

(2) In the revised draft, We added references [24] [37] to prove that “Though former data transaction algorithms have obtained relatively acceptable performance for low dimensional or balanced small-scale datasets.”

(3) We rephrased the sentence “For a long time, the researchers from local institutes tried traditional algorithms, and put them into practice for several years, but these were ineffective.” as “For a long time, researchers from local institutes have tried traditional algorithms and put them into practice for several years [32].”. References [32] are added to prove it.

We excerpt major revisions as follows:

Point 3: In section “3. MATERIALS AND METHODS”

  • Lines 144-146: the authors say they use temperature dataset recorded for more than 15 years but they do not specify the period of time for “the second dataset mainly recorded the precipitation and insects which affects wheat yield.”
  • The subsection 3.2 “Evaluation Indexes and Validation Procedure” has no references although authors state that “Most experts…” and describe different parameters which they are defined in specialized literature.

Response 3: Many thanks for the reviewer’s comment.

(1) We moved the subsection 3.2 to the new subsection 5.1. We rephrased the sentence as “The first one mainly documented the temperature of rice fields in more than 15 years, and the second one mainly recorded the precipitation and insects that have affected rice yield for more than 15 years.”

We specify the period of time of“the second dataset mainly recorded the precipitation and insects which affects wheat yield.” for more than 15 years.

(2) In the subsection 3.2 Evaluation Indexes and Validation Procedure, we added reference [29] for the introduction of assessment indicators.

We excerpt major revisions as follows:

Point 4: In section “4. CPRF ALGORITHM OPTIMIZATION”

  • Subsections 4.1 title makes no sense “Initial of Cascade Random Forest Algorithm”.

Response 4: Many thanks for pointing it out. By viewing the paper carefully, we found this sentence is nonsense,either. So we decide to delete this sentence.

We delete the title of subsection 4.1, and put the content of subsection 4.1 into the subsection 4.2. Then, formed the new subsection 4.1 Cascade Random Forest Algorithm.

We excerpt major revisions as follows:

Point 5:  In section “5. CPRF ALGORITHM OPTIMIZATION”

  • The authors failed to provide a formula for accuracy values in Table 2.
  • Line 464: “RYCC is to obtain the accuracy of the two categories..” should be rephrased.
  • Figure 9 presents several agricultural disease data like Rice planthopper, Rice Leaf Roller, Wheat Stripe Rust which are nowhere mentioned in the text previously.
  • The results in Figure 10 cannot be verified since there is no formula or explanation for the accuracy. So far, from everything authors presented, it is difficult to believe that their CPRF implementation leads to an almost 100% accuracy.
  • One main disadvantage of this work is that it is not comparing the results with the already existing ones in literature. Therefore, the presented results are questionable.

Response 5: We thank the reviewer for the constructive comment. We switched the order of the subsections, moving the accuracy subsection to the back, and the former Table2 became new Table3.

(1) We added formula(12) for accuracy value in the former Table2. We excerpt major revisions as follows:

(2) The former sentence “RYCC is to obtain the accuracy of the two categories.” is rephrased as “The metric RYCC is employed to evaluate the classification accuracy according to the observed classes and predicted classes in test dataset”

(3) In Figure 9, for the dataset not mentioned in the previous section, we removed the experimental results for this part of the dataset.

(4) Figure 10 from the original paper was removed because of the redundancy between the experimental results presented in this figure and the results in Table 3.

Besides, from subsection 5.4 to subsection 5.8 of Experiment results analysis, the method in this paper outperforms the three methods compared in terms of average train_time, average execution time, efficiency and accuracy of its algorithm execution. Because this cascade parallel optimization approach achieves higher efficiency, a less execution time and better workload balance than former algorithms, makes prediction accuracy of CPRF algorithm higher than former algorithms.

The accuracy metric in the former Figure 10 is the RYCC introduced in Equation 11, and CPRF can obtain a higher accuracy because of the introduction of the stacking structure, which enables better feature extraction.

(5) The analysis of the experimental results compared with accuracy of the already existing references has been added to the paper in Table 4. The prediction accuracy of the four different method in the literature [37]- [40] for the dataset of this paper is presented in Table 4. It shows that CPRF has the highest accuracy.

 We excerpt major revisions as picture of the word version in attachment.

Reviewer 3 Report

A random forest predicting model for rice disease detection had been proposed. Several revisions are essential to improve the manuscript. Although the paper has appropriate length and informative content, several parts must be improved and written in better grammar and syntax. It would be essential if authors would consider revising the organization and composition of the manuscript, in terms of the definition/justification of the objectives, description of the method, the accomplishment of the objective, and results.

The paper is generally difficult to follow. Paragraphs and sentences are not well connected. Furthermore, I advise considering using standard keywords to better present the research. update the keywords on methods, and use the standard keywords.

Please revise the abstract according to the journal guideline. The research question, method, and the results must be briefly communicated. The abstract must be more informative.

I suggest having four paragraphs in the introduction for; describing the concept, research gap, contribution, and the organization of the paper. The motivation has the potential to be more elaborated. You may add materials on why doing this research is essential, and what this article would add to the current knowledge, etc. The originality of the paper is not discussed well. The research question must be clearly given in the introduction, in addition to some words on the testable hypothesis. Please elaborate on the importance of this work. Please discuss if the paper suitable for broad international interest and applications or better suited for the local application? Elaborate and discuss this in the introduction.

The reference list cover the relevant literature adequately and in an unbiased manner, however, it needs improvement.

The statistical methods are valid and correctly applied. However, all equations must be numbered and cited.

State of the art needs improvement. A detailed description of the cited references is essential. Several recently published papers are not included in the review section. In fact, the acknowledgment of the past related work by others, in the reference list, is not sufficient. Consequently, the contribution of the paper is not clear. Furthermore, consider elaborating on the suitability of the paper and relevance to the journal. Kindly note that references cited must be up to date, and study further relevant research can improve the state of the art, e.g., A combined method of image processing and artificial neural network for the identification of 13 Iranian rice cultivars; or, Modeling pan evaporation using Gaussian process regression K-nearest neighbors random forest and support vector machines; comparative analysis.

Elaborate on the method used and why used this method.

Limitations and validation are not discussed adequately. The research question and hypothesis must be answered and discussed clearly in the discussion and conclusions. Please communicate the future research. The lessons learned must be further elaborated in the conclusion by discussing the results to the community and the future impacts. What is your perspective on future research?   

Author Response

Dear Reviewer,

We would like to thank the reviewer for providing very insightful comments that helped us improve the quality and completeness of this manuscript. We have carefully considered all your comments and reflected them to prepare this revision. In the following responses, we transcribe the reviewers’ comments and provide point-by-point responses.

In this response letter, we label the textual responses via red font. The content in revised manuscript is labeled as black font or be shown as a screenshot. The key content is labeled via yellow background.

Please download the response letter in this attachment.

Kindly see below for detailed responses.

Sincerely Yours,

Authors

Response to Reviewer Comments

Point 1: A random forest predicting model for rice disease detection had been proposed. Several revisions are essential to improve the manuscript. Although the paper has appropriate length and informative content, several parts must be improved and written in better grammar and syntax. It would be essential if authors would consider revising the organization and composition of the manuscript, in terms of the definition/justification of the objectives, description of the method, the accomplishment of the objective, and results.

Response 1: We thank the reviewer for the constructive comment.

We have placed former subsection 3.1-3.3 into Chapter 5, and the revised article structure basically follows the definition/justification of the objectives, description of the method, the accomplishment of the objective, and results. The details are shown in the table below.

Adviced organizations

Revised manuscript

definition/justification of the objectives,

3.1 Problem formulation

description of the method

Subsection 3.2-4.2

the accomplishment of the objective, and results

5. Experiments results and analysis

Point 2: The paper is generally difficult to follow. Paragraphs and sentences are not well connected. Furthermore, I advise considering using standard keywords to better present the research. update the keywords on methods, and use the standard keywords.

Response 2: Many thanks for pointing it out.

We found some irregular professional words used in the original text and revised them, specifically shown in the table below. 

Original words

New words after revision

random under sampling

random down sampling

data transaction algorithms

data processing algorithms

Point 3: Please revise the abstract according to the journal guideline. The research question, method, and the results must be briefly communicated. The abstract must be more informative.

Response 3: Many thanks for the reviewer’s comment. According to the journal guideline, the abstract was revised as below.

(1) Background (The research question): Experts in agriculture have conducted considerable work on rice plant protection. However, in-depth exploration of the plant disease problem has not been performed. (2) Method: In this paper, we find the trend of rice diseases by using the cascade parallel random forest (CPRF) algorithm on the basis of relevant data analysis in the recent 20 years. To confront the problems of high dimensions and imbalanced data distributions in agricultural data.The proposed method diminishs the dimensions and the negative effect of imbalanced data by cascading several random forests.. For experimenal evaluation, we utilize the Spark platform to analyze Botanic data from several provinces of China in the past 20 years. (3) Results: Results for the CPRF model of plant diseases that affect rice yield, as well as results for samples by using random forest, CRF, and Spark-MLRF are presented, and the accuracy of CPRF is higher than that of the other algorithms. (4) Conclusions: These results indicate that the CPRF and the utilization of big data analysis are beneficial in solving the problem of plant diseases.

  We excerpt major revisions as picture in the attachment.

Point 4: I suggest having four paragraphs in the introduction for; describing the concept, research gap, contribution, and the organization of the paper. The motivation has the potential to be more elaborated. You may add materials on why doing this research is essential, and what this article would add to the current knowledge, etc. The originality of the paper is not discussed well. The research question must be clearly given in the introduction, in addition to some words on the testable hypothesis. Please elaborate on the importance of this work. Please discuss if the paper suitable for broad international interest and applications or better suited for the local application? Elaborate and discuss this in the introduction.

Response 4: We thank the reviewer for the constructive comment.

We have reorganized the structure of the Introduction as follows.

1.1 Research Background and meaning

1.2 Research gap

1.3 contribution

1.4. Organizations of the paper

In subsection 1.1, We introduced the research background and meaning.

The yield of agricultural products is considerably important for every country. Although a nation has abundant agricultural product resources, agricultural plant protection and botany disease prevention should be considered seriously. For efficient management of crop farming, the processing of such data as rice disease-affected area, temperature, and precipitation is important for the effectiveness of an algorithm.

In subsection 3.1, we give the symbolic definition and the formal formulation of the proposed problem in this paper.

 We excerpt major revisions as picture in the attachment

We discuss the paper is better suited for the local application, because the method in this paper is suitable for partial application in local agriculture, and we haven’t made other experiments in other field because we lack of their original data, the corresponding explanation is added in the conclusion.

 We excerpt major revisions as picture in the attachment

Point 5: The reference list cover the relevant literature adequately and in an unbiased manner, however, it needs improvement.

Response 5: Many thanks for the reviewer’s comment.

The related work subsection of the paper was revised thoroughly. At the same time, the references are revised according to the newly related work.

The excerpts of references list are shown as follows:

References

  1. Yousef Abbaspour-Gilandeh, et al. A combined method of image processing and artificial neural network for the identification of 13 Iranian rice cultivars-agronomy[J]. agronomy Vol 10(2020), pp. 1-21.
  2. Everingham, Y, et al. Accurate prediction of sugarcane yield using a random forest algorithm [J]. AGRONOMY FOR SUSTAINABLE DEVELOPMENT Vol 36 (2016), pp. 1-9.
  3. Zheng Chu, Jiong Yu. An end-to-end model for rice yield prediction using deep learning fusion [J]. Computers and Electronics in Agriculture 174 (2020) 105471, pp. 1-11
  4. LiTian, et al. Yield prediction model of rice and wheat crops based on ecological distance algorithm [J]. Environmental Technology & Innovation Vol 20 (2020), pp. 1-12.
  5. Barzin, R, et al. Use of UAS Multispectral Imagery at Different Physiological Stages for Yield Prediction and Input Resource Optimization in Corn [J]. REMOTE SENSING Vol 12(2020), pp. 1-21.
  6. Rashid, M, et al. A Comprehensive Review of Crop Yield Prediction Using Machine Learning Approaches With Special Emphasis on Palm Oil Yield Prediction[J]. IEEE ACCESS Vol 9 (2021), pp. 63406-63439.
  7. Peng, B, et al. Assessing the benefit of satellite-based Solar-Induced Chlorophyll Fluorescence in crop yield prediction [J]. INTERNATIONAL JOURNAL OF APPLIED EARTH OBSERVATION AND GEOINFORMATION Vol 90 (2020), pp. 1-15.
  8. Ju-Young, S, et al. Seasonal forecasting of daily mean air temperatures using a coupled global climate model and machine learning algorithm for field-scale agricultural management [J]. AGRICULTURAL AND FOREST METEOROLOGY Vol 281 (2020), pp. 1-16.
  9. Khosla, E , et al. Crop yield prediction using aggregated rainfall-based modular artificial neural networks and support vector regression [J]. ENVIRONMENT DEVELOPMENT AND SUSTAINABILITY Vol 22 (2020), pp. 5687-5708.
  10. Vimala, et al. Optimal Routing and Deep Regression Neural Network for Rice Leaf Disease Prediction in IoT [J]. International Journal of Computational Methods Vol 18, No. 07, 2150014 (2021) , pp. 1-10.
  11. van Kloppenburg, et al. Crop yield prediction using machine learning: A systematic literature review [J]. Computers and Electronics in Agriculture 177 (2020) 105709, pp. 1-18.
  12. Chen, S, et al. Dynamic within-season irrigation scheduling for maize production in Northwest China A Method Based on Weather Data Fusion and yield prediction by DSSAT [J]. Agricultural and Forest Meteorology Vol 285 (2020), pp. 1-23.
  13. Veerakachen, W, et al. RiceSAP: An Efficient Satellite-Based AquaCrop Platform for Rice Crop Monitoring and Yield Prediction on a Farm- to Regional-Scale[J]. AGRONOMY-BASEL Vol 10(2020), pp. 1-17.
  14. Sharifi, A, et al. Yield prediction with machine learning algorithms and satellite images [J]. JOURNAL OF THE SCIENCE OF FOOD AND AGRICULTURE Vol 101 (2020), pp. 891-896.
  15. Sun, SP, et al. Image processing algorithms for infield single cotton boll counting and yield prediction [J]. COMPUTERS AND ELECTRONICS IN AGRICULTURE Vol 166(2019), pp. 1-15.
  16. Das, S, et al. Evaluation of water status of genotypes to aid prediction of yield on sodic soils using UAV-thermal imaging and machine learning [J]. AGRICULTURAL AND FOREST METEOROLOGY Vol 307 (2021), 0168-1923. pp.1-15.
  17. Esfandiarpour-Boroujeni, I, et al. Yield prediction of apricot using a hybrid particle swarm optimization-imperialist competitive algorithm- support vector regression (PSO-ICA-SVR) method [J]. SCIENTIA HORTICULTURAE Vol 257 (2019), pp. 1-12.
  18. Jose Clodoalves, et al. Random forest techniques for spatial interpolation of evapotranspiration data from Brazilian’s Northeast [J]. Computers and Electronics in Agriculture 166 (2019) 105017, pp. 1-10.
  19. Mariano, C, et al. A random forest-based algorithm for data-intensive spatial interpolation in crop yield mapping. [J]. COMPUTERS AND ELECTRONICS IN AGRICULTURE Vol 184 (May 2021), pp.1-9.
  20. Zhi-Sen Wei, et al. A Cascade Random Forests Algorithm for Predicting Protein-Protein Interaction Sites [J]. IEEE TRANSACTIONS ON NANOBIOSCIENCE, VOL. 14, NO. 7, OCTOBER 2015, pp.746–760.
  21. Jianguo Chen, et al. A Parallel Random Forest Algorithm for Big Data in a Spark Cloud Computing Environment [J]. IEEE Transactions on Parallel and Distributed Systems, VOL. 28, NO. 4, APRIL 2017, pp.919–933.
  22. da Silva, JC, et al. Random forest techniques for spatial interpolation of evapotranspiration data from Brazilian's Northeast [J]. COMPUTERS AND ELECTRONICS IN AGRICULTURE Vol 166 (2019), pp. 1-10.
  23. Sevda Shabani, et al. Modeling pan evaporation using Gaussian process regression K-nearest neighbors random forest and support vector machines [J]. atmosphere, 2020, 11, 66, pp. 1-17.
  24. Lintas, A. et al. ReForeSt: Random Forests in Apache Spark [J]. ARTIFICIAL NEURAL NETWORKS AND MACHINE LEARNING, PT II 10614(2017). pp. 331-339.
  25. Ahmed, AAM, et al. Deep learning hybrid model with Boruta-Random forest optimiser algorithm for streamflow forecasting with climate mode indices, rainfall, and periodicity [J]. JOURNAL OF HYDROLOGY Vol 599 (2021), pp. 1-23.
  26. WeiWei Lin, et al. An Ensemble Random Forest Algorithm for Insurance Big Data Analysis[J]. IEEE Access. 10.1109/ACCESS.2017.2738069.Volume 5, 2017,pp.16568-16575.
  27. Xu, H. Chen, and P. K. Varshney, Dimensionality reduction for registration of high-dimensional data sets [J]. IEEE Trans. Image Process., vol. 22, no. 8, Aug. 2013, pp. 3041–3049.
  28. Wang, H, et al. Integrating remotely sensed leaf area index and leaf nitrogen accumulation with RiceGrow model based on particle swarm optimization algorithm for rice grain yield assessment [J]. JOURNAL OF APPLIED REMOTE SENSING Vol 8 (2014), pp. 083674-1-16.
  29. Paudel, Dilli, et al. Machine learning for large-scale crop yield forecasting [J]. AGRICULTURAL SYSTEMS Vol 187 (2020), pp. 1-13.
  30. Romeiko, X X, et al.Comparing Machine Learning Approaches for Predicting Spatially Explicit Life Cycle Global Warming and Eutrophication Impacts from Corn Production [J]. SUSTAINABILITY Vol 12 (2020), pp. 1-19.
  31. Kang, YH, et al. Comparative assessment of environmental variables and machine learning algorithms for maize yield prediction in the US Midwest [J]. ENVIRONMENTAL RESEARCH LETTERS Vol 15 (2020), pp. 1-12.
  32. Puyu Feng, et al. Dynamic wheat yield forecasts are improved by a hybrid approach using a biophysical model and machine learning technique [J]. Agricultural and Forest Meteorology Vol 285-286 (2020), pp. 1-12.
  33. Grace, RK, et al. Enrichment of Crop Yield Prophecy Using Machine Learning Algorithms [J]. INTELLIGENT AUTOMATION AND SOFT COMPUTING Vol 31 (2022), pp. 279-296.
  34. Wen, GQ, et al. Machine learning-based canola yield prediction for site-specific nitrogen recommendations [J]. NUTRIENT CYCLING IN AGROECOSYSTEMS Vol 121 (2021), pp. 241-256.
  35. Folberth, C, et al. Spatio-temporal downscaling of gridded crop model yield estimates based on machine learning [J]. AGRICULTURAL AND FOREST METEOROLOGY Vol 264 (2019), pp. 1-15.
  36. Khanal, S, et al. Assessing the impact of agricultural field traffic on corn grain yield using remote sensing and machine learning [J]. SOIL & TILLAGE RESEARCH Vol 208 (2021), pp. 1-11.
  37. Ze He, et al. MAPPING RICE PLANTING AREA USING MULTI-TEMPORAL QUAD-POL RADARSAT-2 DATASETS AND RANDOM FOREST ALGORITHM [J]. IEEE International Geoscience and Remote Sensing Symposium. OCT 02, 2020, pp. 4653-4656.
  38. Seok-Jun Bu, et al. Ensemble of Deep Convolutional Learning Classifier System Based on Genetic Algorithm for Database Intrusion Detect [J]. Electronics Vol 11, 745,2022, pp. 1-16.
  39. Samiul Alam, et al. BORO RICE YIELD ESTIMATION MODEL USING MODIS NDVI DATA FOR BANGLADESH [J]. IEEE International Geoscience and Remote Sensing Symposium . AUG 02, 2019, pp. 7330-7333.
  40. Kazi A Kalpoma, Ashiqur Rahman. WEB-BASED MONITORING OF BORO RICE PRODUCTION USING IMPROVISED NDVI THRESHOLD OF MODIS MOD13Q1 AND MYD13Q1 IMAGES [J]. IEEE International Geoscience and Remote Sensing Symposium 2021, pp. 6877-6880.

Point 6: The statistical methods are valid and correctly applied. However, all equations must be numbered and cited.

Response 6: Many thanks for pointing it out. Your comments are very important for authors.

We checked all statistical indicators and made sure that each indicator was numbered and referenced.

In the original paper, we forgot to number the equation (11). We revised the paper, and added the new equation (12).

 We excerpt major revisions as picture in the attachment

Point 7: State of the art needs improvement. A detailed description of the cited references is essential. Several recently published papers are not included in the review section. In fact, the acknowledgment of the past related work by others, in the reference list, is not sufficient. Consequently, the contribution of the paper is not clear. Furthermore, consider elaborating on the suitability of the paper and relevance to the journal. Kindly note that references cited must be up to date, and study further relevant research can improve the state of the art, e.g., A combined method of image processing and artificial neural network for the identification of 13 Iranian rice cultivars; or, Modeling pan evaporation using Gaussian process regression K-nearest neighbors random forest and support vector machines; comparative analysis.

Response 7: Many thanks for pointing it out. The two papers from the journals agronomy and atmosphere respectively are excellent. We have cited the two papers recommended by you, and on this basis, we have cited several relatively new research works in the field, Specifically including the literature cited in reference [1] and [23].

 We excerpt major revisions as picture in the attachment

Point 8: Elaborate on the method used and why used this method.

Response 8: Many thanks for the reviewer’s comment.

In the subsection 3.3 CPRF, we supplied the necessity of introducing CPRF algorithm:

The excerpts of references list are shown as follows:

In section 4 CPRF Algorithm Optimization, in the first paragraph we added the necessity to use CPRF.

With the continual improvement in agricultural prediction accuracy for complicated and large-volume data, an optimization method for the CPRF algorithm is proposed. First, imbalance and dimension reduction approaches [20] are performed in the training process. Second, random under sampling and balanced training are implemented. Then, several minority classes are obtained. For example, pest’s activity scope is a kind of minority class, which affects the harvest of agricultural plants. After regression and classification optimizations, the prediction accuracy of the algorithm is evidently improved.

Point 9: Limitations and validation are not discussed adequately. The research question and hypothesis must be answered and discussed clearly in the discussion and conclusions. Please communicate the future research. The lessons learned must be further elaborated in the conclusion by discussing the results to the community and the future impacts. What is your perspective on future research?   

Response 9: Many thanks for the reviewer’s comment.

In the conclusion section, we add the limitations of the method in this paper and the prospect of future work. The limitation of this paper is that its model training process relies on offline data training, and the data needs to be collected and pre-processed manually. Our perspective on future research is the original data collection could be optimized with semi-supervised learning,

The revised conclusion is shown in the word version in the attachment.

Round 2

Reviewer 1 Report

I have no more suggestions!

Reviewer 2 Report

The authors have significantly improved their work following round 1 of revisions. Two observations must be addressed: the references numbering should be made consecutively (e.g. after 37 there can not be 32); since section 4 has only one subsection (4.1) I suggest that you eliminate this numbering.

Reviewer 3 Report

All the comments had been addressed.